# Anatomy and Imaging of Rat Prostate: Practical Monitoring in Experimental Cancer-Induced Protocols

**DOI:** 10.3390/diagnostics9030068

**Published:** 2019-06-30

**Authors:** Mário Ginja, Maria J. Pires, José M. Gonzalo-Orden, Fernanda Seixas, Miguel Correia-Cardoso, Rita Ferreira, Margarida Fardilha, Paula A. Oliveira, Ana I. Faustino-Rocha

**Affiliations:** 1Department of Veterinary Sciences, University of Trás-os-Montes and Alto Douro (UTAD), 5001-801 Vila Real, Portugal; 2Center for the Research and Technology of Agro-Environmental and Biological Sciences (CITAB), 5001-801 Vila Real, Portugal; 3Department of Medicine, Surgery and Veterinary Anatomy, University of León, 24071 León, Spain; 4Animal and Veterinary Research Center (CECAV), 5001-801 Vila Real, Portugal; 5Organic Chemistry, Natural Products and Food Stuffs (QOPNA), 3810-193 Aveiro, Portugal; 6Department of Medical Sciences, University of Aveiro, 3810-193 Aveiro, Portugal; 7Faculty of Veterinary Medicine, Lusophone University of Humanities and Technologies (ULHT), 1749-024 Lisbon, Portugal

**Keywords:** computed tomography (CT), macroscopy, magnetic resonance imaging (MRI), microscopy, ultrasonography

## Abstract

The rat has been frequently used as a model to study several human diseases, including cancer. In many research protocols using cancer models, researchers find it difficult to perform several of the most commonly used techniques and to compare their results. Although the protocols for the study of carcinogenesis are based on the macroscopic and microscopic anatomy of organs, few studies focus on the use of imaging. The use of imaging modalities to monitor the development of cancer avoids the need for intermediate sacrifice to assess the status of induced lesions, thus reducing the number of animals used in experiments. Our work intends to provide a complete and systematic overview of rat prostate anatomy and imaging, facilitating the monitoring of prostate cancer development through different imaging modalities, such as ultrasonography, computed tomography (CT) and magnetic resonance imaging (MRI).

## 1. Introduction

The prostate is the largest accessory gland of the male reproductive tract [1], and is responsible for the secretion of a slightly alkaline fluid that forms part of the seminal fluid [2]. This accessory gland is not exclusive to males, with it also being present in female Mongolian gerbils [3].

Prostate cancer is the second most common cancer in men. In 2012, it affected approximately 1.1 million men and caused 307,000 deaths worldwide [4]. Prostate cancer can be studied in vitro, allowing for the understanding of biological mechanisms underlying the development of the disease, but the current in vitro models fail to mimic the cell interactions that occurs in the tumor microenvironment. These data mean that animal models are of great importance for the study of prostate carcinogenesis, and for the development of new pharmacological and non-pharmacological prophylactic and therapeutic strategies to fight this disease. Several animal models are currently available for the study of prostate cancer carcinogenesis, including: spontaneous, chemically or hormonally induced, implantation of cancer cells and genetically engineered animals [5,6,7].

The rat was first used as a model of prostate cancer in 1937 by Moore and Melchionna [8]. Since the anatomy of the prostate varies greatly between species, a document providing researchers with a complete overview of the macroscopic and microscopic anatomy of the rat prostate, and its non-invasive monitoring by different imaging modalities, such as ultrasonography, computed tomography (CT), magnetic resonance imaging (MRI) and positron emission tomography (PET) is sorely needed.

This work intends to fill this gap, describing the normal and abnormal cancer macroscopic and microscopic anatomy of the rat prostate gland and its imaging monitoring by ultrasonography, CT and MRI.

## 2. Prostate Cancer Induction Protocol

The images included in this article are part of an experimental assay aiming to characterize prostate cancer evolution in male Wistar Unilever (*Rattus norvegicus*) rats. All procedures were performed according to the European Directive 2010/63/EU on the protection of animals used for scientific purposes. The Portuguese Ethics Committee for Animal Experimentation (*Direcção Geral Alimentação e Veterinária*) approved all of the experiments and procedures carried out on the animals (ethical approval no. 021326, 26th October 2016), after the approval (427-e-DCV-2016) from the committee for animal welfare of the University of Trás-os-Montes and Alto Douro (18th April 2016).

The prostates of five rats (two normal and three with cancer induction) between 17 to 61 weeks of age were evaluated by ultrasonography, CT and MRI. The animals were maintained under controlled conditions, with stable temperature (23 ± 2°C), humidity (50 ± 10%), air system filtration (10–20 ventilations/hour) and light:dark cycle (12-h:12-h). They had free access to water and a standard laboratory diet (Mucedola 4RF21^®^, Milan, Italy). The water was changed and the cages were cleaned weekly.

To induce prostate cancer the following protocol was performed in three animals: At 12 weeks of age, the animals from the prostate cancer group received a subcutaneous administration of the anti-androgenic drug flutamide (50 mg/kg of body weight; TCI Chemicals, Portland, OR, USA) for 21 consecutive days. Twenty-four hours after the last administration of flutamide, testosterone propionate (TCI Chemicals, Portland, OR, USA) was dissolved in corn oil and subcutaneously administered to the animals at a dose of 100 mg/kg. Forty-eight hours later, the animals were intraperitoneally injected with the carcinogenic agent *N*-Methyl-*N*-nitrosourea (MNU) (Isopac^®^, Sigma Chemical Co., Madrid, Spain), at a dose of 30 mg/kg. Two weeks later, silastic tubes filled with crystalline testosterone (Sigma Chemical Co., Madrid, Spain) were subcutaneously implanted for a maximum period of 44 weeks in the interscapular region of animals previously anesthetized with ketamine (75 mg/kg, Imalgene^®^ 1000, Merial S.A.S., Lyon, France) and xylazine (10 mg/kg, Rompun^®^ 2%, Bayer Healthcare S.A., Kiel, Germany). The two control animals were not subjected to any treatment. At the end of the experiment, all animals were sacrificed through an intraperitoneal injection of ketamine (75 mg/kg, Imalgene^®^ 1000, Merial S.A.S., Lyon, France) and xylazine (10 mg/kg, Rompun^®^ 2%, Bayer Healthcare S.A., Kiel, Germany), followed by exsanguination by cardiac puncture. The prostate from each animal was then collected and processed for histological analysis.

## 3. Macroscopic Anatomy

The anatomy of the prostate gland varies widely between species, but it is generally located below the bladder and in front of the rectum [5,6,7,8,9]. The prostate is a compact structure in men and dogs, but it is composed of several lobes in rats and mice. The rat prostate is composed of four distinct lobes with different morphological characteristics which are commonly called the ventral, lateral, dorsal and anterior lobes, and are classified according to their relative position to the urinary bladder [5,6,7,8,9] (Figure 1). Despite the anatomical differences, prostate carcinogenesis in rats and men is controlled by similar molecular mechanisms, making the rat a valuable model for the study of human prostate diseases, including cancer [7].

## 4. Microscopic Anatomy

The rat prostate is a highly specialized tubulo-alveolar exocrine gland [10]. It consists of four distinct paired (right and left) lobes, the dorsal prostate gland, lateral prostate gland, ventral prostate gland, and anterior prostate gland or coagulating gland [11,12,13], classified according to their position in relation to the urethra, into which the glandular ducts drain (Figure 2A) [10,14]. Each of the prostate lobes has a distinctive histology. For the evaluation of histopathological lesions, it is important that each gland is fully represented and correctly identified [12]. Reliable identification of the lobes depends on obtaining a cut section that maintains the anatomical relationships of each lobe to the others and to the urethra [12,15].

Histologically, the prostate lobes are surrounded by a thin mesothelial connective capsule. Each lobe is composed of individual glands (alveoli or acini) and a series of branching ducts that drain independently into the urethra [16]. The acini are separated by a thin loose connective tissue that contains stromal cells, interspersed smooth muscle cells, vessels, nerves, ganglia [13], macrophages, and mast cells [17]. The cells lining the acini and ducts include luminal secretory cells, non-secretory basal cells (less frequent, corresponding to 2% of the acinar cells) [17], and a low number of neuroendocrine cells [18]. The luminal cells vary from cuboid to tall columnar and the height of the cells depends upon the degree of secretory activity and glandular distension [12,19]. The acini are surrounded by smooth muscle that contract to expel the prostate secretions [17]; depending on the secretory activity, the lobes are filled with a proteinaceous secretion that stains slightly eosinophilic [12].

The dorsal prostate is located below and behind the attachment of the seminal vesicle and the coagulating gland and surrounds the urethra dorsally (Figure 2B) [11]. Microscopically, it resembles the coagulating gland [12,17]. The acini are small with reduced infolding of the epithelium [15,18] and are loosely distributed within the stroma [11,15,18]. The acinar lining cells are generally cuboid [15], but may vary from cuboidal to tall columnar, with cytoplasmic blebs (Figure 2C and Figure 2D) [13,17], and centrally located nuclei [13]. The acini are surrounded by a thin fibromuscular layer [13,17], and contain a slightly eosinophilic secretion [11,18] that stains an intermediate color between that of the secretions of the lateral and ventral lobes [12].

The lateral prostate is just below the seminal vesicle and coagulating gland [11] and is lined by a simple cuboid [11,13] to tall columnar epithelium [13], with basal nuclei and few to moderate areas of epithelial infolding (Figure 2E) [11,13]. These cells contain an eosinophilic cytoplasm that is less granular than that of the dorsal prostate, and which has a distinctive brush border [12] with prominent supranuclear areas of pallor (Figure 2F) [15]. The glandular lumen may be of different sizes, from small to large, and contains a strongly eosinophilic secretion [10,12,13,15,17,20]. Due to the relative difficulty associated with the anatomic individualization of the dorsal and lateral lobes [10,18], and similar overlapping histological features [17], these lobes are usually classified as a single element, called the dorsolateral prostate.

The ventral prostate arises from the ventral aspect of the urethra, immediately below the bladder [11]. It is the largest lobe, constituting approximately half of the mass of all the prostatic tissue, and is most easily separated from the rest of the prostate [11]. It is composed of closely packed varying-sized acini lined by low to tall columnar epithelium [11,13], basophilic cytoplasm (Figure 2G) [15,19], a basally located nucleus and a supranuclear clear area (Figure 2H) [11,13,15]. Glands show very scarce infolding [11,13,15,19], are surrounded by a very small amount of smooth muscle, and contain pale eosinophilic serous [12,13,15,18] or flocculent secretions [12,19].

The coagulating gland is sometimes referred as the dorsocranial, cranial, or anterior prostate [12,20], and lies adjacent and parallel to the concave surface of the seminal vesicle [11,12]. The acini are tightly packed and surrounded by a prominent fibromuscular layer [15]. The glands are lined by simple cuboid to columnar epithelium [11,12,13], with an eosinophilic granular cytoplasm, a centrally located nucleus and an inconspicuous nucleolus [13]. The cells are arranged in extensive branching pattern, forming papillary, or cribriform structures [12,13,15], and the lumen contains an abundant, homogeneous, pale eosinophilic proteinaceous secretion (Figure 2I and Figure 2J) [12,13] similar to that seen in the dorsal prostate [12]. The epithelial height varies throughout the gland, as does the secretory activity [11].

## 5. Prostate Imaging

Prostate cancer-related deaths have decreased over the last few years, in part due to the extensive use of screening strategies such as digital rectal examination and the measurement of serum levels of prostate-specific antigen (PSA) [21]. Furthermore, the prostate gland and prostate carcinogenesis may be non-invasively monitored through different imaging modalities, such as ultrasonography, CT, MRI, and PET/CT.

### 5.1. Ultrasonography

Ultrasonography is the oldest and most widely used technique for anatomical imaging in clinical practice [21]. It has some advantages when compared with other imaging modalities: it does not impose radiation that may harm the patient or the handler [22], it allows for dynamic and real-time study, may be used guiding prostate biopsies, it is better tolerated than other imaging modalities in patients with claustrophobia, it can be used in patients with pacemakers or other metal implants, and it is portable and less expensive than most other imaging modalities [22,23,24,25,26]. However, ultrasonography also has several important clinical limitations in differential diagnosis of cancerous tissue and nonmalignant conditions and low sensitivity in early diagnosis [26]. In terms of experimental research, where the animals are monitored several times throughout the experiment, the fact that ultrasonography can be performed in awake animals previously adapted to the researchers constitutes an additional advantage when compared with CT and MRI, in which the animals need to be anesthetized beforehand. Ultrasonography allows not only for the study of prostate anatomy (dimension, shape, structure of parenchyma), but can also evaluate the tumor microenvironment (prostate vascularization, pattern of distribution of the vessels inside the parenchyma) through the use of distinct modes, like B mode, Power Doppler, Color Doppler, Pulsed Doppler, B Flow and Contrast-enhanced ultrasound [26,27].

*Procedure for rat prostate monitoring:* For ultrasonographic examination, alert animals should be restrained by a researcher and placed in the supine position. The skin of the caudal aspect of the abdomen and the inguinal region should be shaved using a machine clipper (AESCULAP^®^ GT420 Isis, Aesculap Inc, Center Valley, PA, USA) (Figure 3A). A real-time scanner (Logiq P6^®^, General Electric Healthcare, Milwaukee, WI, USA) and a 12 MHz linear transducer may be used, and acoustic gel needs to be applied (Aquasonic, Parker Laboratories Inc., Fairfield, NJ, USA). A complete transverse scan from the caudal aspect of the abdomen to the inguinal region (Figure 3B), and a longitudinal scan (Figure 3C) should be performed using B mode.

Transverse scan: In the transverse scan, the urinary bladder presents as a round to oval shape filled with urine (anechoic structure) and the prostate lobes are visible around the urinary bladder. The ventral prostate lobes appear as hypoechoic elongated structures (one right and one left) with a hyperechoic capsule, placed ventrally to the urinary bladder (Figure 4A, B). In this scan, the dorsal prostate is only observed close to the urinary bladder neck (urinary bladder with low diameter, almost disappearing from the screen) (Figure 4B). The dorsal prostate appears as a round hypoechoic structure with a hyperechoic capsule, placed dorsally to the urinary bladder. The seminal vesicles (accessory glands of the male genital tract) are also observed in this transverse scan. They are seen as the hypoechoic elongated structures (less hypoechoic when compared with prostate gland lobes), placed dorsally to the urinary bladder, in the transition area between the cranial abdomen and urinary bladder neck (Figure 4A).

Sagittal scan: In the sagittal scan, the urinary bladder is observed as an elongated structure filled with urine (anechoic content). The ventral prostate lobes are occasionally observed ventrally to the neck of the urinary bladder, with an appearance as previously described for the transverse scan. The dorsal prostate is observed dorsal to the neck of the urinary bladder, presenting as a round to elongated shape, with a hypoechoic appearance and a hyperechoic capsule (Figure 5).

Monitoring of the prostate during the experimental induction of prostate cancer: The development of prostate cancer was induced through a multistep protocol. The prostates of both the control (Figure 6A–F) and prostate cancer-induced (Figure 6G–L) groups were monitored at different time points during the experimental protocol. The transverse scan allowed for the observation of the ventral prostate lobes and seminal vesicle around the urinary bladder, and the longitudinal scan allowed for the observation of the dorsal prostate. The first scan was performed one week before the beginning of the experiment (11 weeks of age) (Figure 6A,G), and showed similar morphologic characteristics in both groups. The second scan was performed three weeks after the administration of the anti-androgen drug flutamide (15 weeks of age), and a reduction of the size of the ventral prostate lobes was observed in the prostate cancer-induced group (Figure 6H). The third scan was performed six weeks after flutamide administration and four weeks after the administration of the carcinogenic agent MNU (21 weeks of age) and demonstrated an increase of the size of the prostate lobes, when compared to the previous exam (Figure 6I). The fourth US exam was performed 17 weeks after the administration of MNU (32 weeks of age), and an increase in prostate size was also observed (Figure 6J). The last US exam was performed 46 weeks after the administration of MNU and an increase in size was mainly observed in the dorsal prostatic lobe (Figure 6L). In the control group, the size of the prostate lobes gradually increased from the first to the last exam (Figure 6A–F).

Due to the mentioned limitations of ultrasonography for detecting and locating prostate cancer, and in finding the exact boundaries between benign and malignant tissue, other imaging modalities, such CT, MRI and PET/CT are therefore employed [28].

### 5.2. Computed Tomography (CT)

CT has had a massive impact on medical practice over the last 40 years [29]. By creating detailed images of large anatomical regions in few seconds, CT has allowed for a deepened understanding of anatomy, physiology and pathology, with improved diagnosis, a reduction in the number of unneeded medical procedures and increased success of treatment. Despite this, CT has important limitations in soft tissues differentiation (normal and cancerous) and use ionizing radiation with inevitable side effects [29,30], being rarely used for human prostate cancer diagnosis or monitoring.

Procedure for rat prostate monitoring: For CT, the animals should be anesthetized by an intraperitoneal injection of ketamine (75 mg/kg, Imalgene^®^ 1000, Merial S.A.S., Lyon, France) and xylazine (10 mg/kg, Rompun^®^ 2%, Bayer Healthcare S.A., Kiel, Germany). They should be placed in a supine position and a scan can be performed with the Brivo CT325 scanner (General Electric Healthcare, Milwaukee, WI, USA). A 24G catheter (BBraun, Barcarena, Portugal) should be inserted in the caudal vein for the administration of about 1 mL (900 mg I/Kg) of the iodine-based contrast agent Ultravist 300 (Bayer HealthCare, Berlin, Germany) [31,32].

Scan: The urinary bladder presents as a round structure with a lower signal than the prostate lobes and seminal vesicles. The ventral prostate lobe is observed ventral to the urinary bladder, while the seminal vesicles are observed in a dorsal position (Figure 7). The contrast-enhanced images do not allow a better visualization of the prostate lobes, neither become visible prostate zones with distinct contrast enhancement. Prostate cancer, even in early phase is characterized by an increased blood flow due to the angiogenesis [33].

### 5.3. Magnetic Resonance Imaging (MRI)

MRI is a safe diagnostic technique widely used in medicine. Advances in MRI have improved the detection and characterization of prostate cancer due to the use of a multiparametric approach that combines anatomical and functional data [2]. MRI T2-weighted images (WI) sequences allow excellent zonal anatomy of human prostate, a characteristic low-signal intensity of cancerous tissue and detection of disease extension [34]. However, other benign prostate conditions are also characterized by T2 WI low signal intensity and a definitive diagnosis of prostate cancer is not possible [34]. Unlike CT, MRI does not use ionizing radiation and works by measuring the reversal of the atomic spin, and not by changing their structure, composition or properties [35]. However, the relatively high cost, the requirement of injected contrast agents for functional imaging, no detection of calcification, claustrophobia and the long scanning time required constitute the main disadvantages of MRI [36].

Procedure for rat prostate monitoring: The anesthetic protocol described above for CT is also adequate for MRI. The animals should be placed in a supine position, and both axial and sagittal scans can be performed with the GE MRI 3 Tesla scan (General Electric Healthcare, Milwaukee, WI, USA) using a coil encompassing the rat pelvic region.

Scan: Proton-density (PD) (Repetition time (TR) = 1240, Echo time (TE) = 40) and T2 (TR = 4000, TE = 100) weighted studies with transverse plane images (slice thickness of 3 mm) allow for a good visualization of the prostate and surrounding structures (Figure 8A–C). The urinary bladder presents as a round structure surrounded by the ventral prostate and the seminal vesicles. The seminal vesicles exhibit low signal when compared to the prostate and urinary bladder in both PD and T2 images (Figure 8A–C). Morphological differences between the prostate and seminal vesicles in normal and induced rats can be clearly observed. In T1 (TR = 300, TE = 14) weighted studies, the prostate and surrounding structures exhibit a low signal (Figure 8D). Scanning in other planes, such as dorsal and sagittal views, also help to provide good three-dimensional information about the prostate and surrounding tissues (Figure 9).

### 5.4. Positron Emission Tomography/Computed Tomography (PET/CT)

Positron emission tomography/computed tomography (PET/CT) using a radiotracer with glucose is a common procedure in oncologic human diagnosis as it results in differential tissue signal, dependent of high or low glucose tissue metabolism [37]. PET/CT is particularly helpful in early diagnosis, to detect metastases or to assess treatment responses and to provide prognostic indicators [38]. However, this common PET/CT radiotracer showed limitations in detection of primary prostate cancer, due to the low glucose metabolic rate and proximity of urinary excretion [37,39]. Some recent studies using specific radiotracers without glucose, as ^11^C-choline, 18F-fluciclovine, prostate-specific membrane antigen (PSMA) and 18F-sodium fluoride, had promising results, distinguishing areas of prostatic cancer from normal tissue, as well as evaluating bone metastasis [34,40,41]. PET-CT evaluation results in more accurate assessment of patients with recurrent high PSA serum levels and overcome many limitations of other medical imaging modalities [42]. ^11^C-choline targets cell membrane lipids biosynthesis enhanced in cancer cells [40]. 18F-fluciclovine is an amino acid analog with an enhanced uptake in cancer cells [40]. PSMA is normally expressed in epithelial cells within the prostate and several PSMA ligands are commercially available as Gallium-68, Fluorine-18 and Copper-64 [43]. The ^11^C-choline, 18F-fluciclovine and PSMA signal positively correlates with PSA serum levels and had a good sensitivity to find sites of prostatic cancer recurrence after radical prostatectomy or radiation therapy [40,43]. Current evidence suggests that PSMA tracers have the best detection performance with low serum PSA levels [40,42,43]. PSMA is overexpressed in prostate cancer cells being associated with cancer aggressiveness, metastasis and recurrence [33,43]. The 18F-sodium fluoride is useful for evaluating bone metastasis, is highly sensitive to osteoblastic activity and has better image resolution, sensitivity and specificity than conventional bone scintigraphy [40]. PET/CT studies in rat prostate cancer model are scarce; however it has sometimes been used to test new radiotracers or to study tissue hypoxia [44,45].

## 6. Discussion

Considering the high incidence and the high mortality of prostate cancer, its study is of great importance. Researchers have therefore employed their efforts to develop in vivo models, which are of paramount importance for the comprehension of this disease and for the development of new preventive and therapeutic strategies to fight it. Rats are frequently considered to be the best model for the study of various human diseases, because they are relatively cheap when compared with other species, their physiology and genetics are well-known, they are easy to manipulate, the carcinogenesis process occurs fast (initiation, promotion, progression and metastasis may be observed) and most importantly they are mammals like humans [46,47]. Although the rat has been frequently used as a model to study prostate disease, including prostate cancer, there are few reports on the normal anatomy of the rat prostate and its monitoring by different imaging modalities, such as ultrasound, CT, MRI or PET.

Nowadays, imaging is considered indispensable in clinical management of human prostate cancer being helpful for abnormal tissue detection, localization or guiding needle biopsies [34,48]. An important milestone was the introduction of ultrasonography to identify lesions and guide transrectal biopsies in 1989 in humans [48]. However, in the meantime, other imaging modalities have emerged with advantages in terms of precocity of diagnosis and delimitation of lesions [34]. The rat was firstly used for the study of prostate carcinogenesis in 1937 by Moore and Melchionna [8]. Prostate cancer induction protocol leads to the enlargement of prostate rat ventral and dorsal lobes, as well seminal vesicles. This enlargement is well detected by ultrasonography, MRI or CT. In rat the MRI seems to be better than ultrasonography to detect and localize abnormal prostate and seminal vesicles tissue (Figure 10). The appearance of rat prostate dorsal and ventral lobes in MRI sequences is similar being evident a homogeneous high-intensity signal in T2 and PD WI. Normal seminal rat vesicles in MRI T2 and PD WI also show a high-intensity signal, but its enlargement was associated with MRI signal loss. The seminal vesicles enlarged showed mural atrophy, flattened papillary folds, reduction of glandular structures, atrophy of muscle layer and in some areas, inside the glandular structures, we observed small masses of eosinophilic laminated concretions of luminal secretions, variable in size and shape (corpora amylacea) (Figure 11A). The human prostate peripherical zone in MRI T2 weighted sequences is also characterized by a high-intensity signal due to water rich tissue composed by numerous ductal and acinar elements [34]. In this zone, cancerous lesions appear as low-intensity signal ill-defined tissue. However, these low-intensity changes are also characteristic of inflammatory conditions, atrophy and local hemorrhages and definitive diagnosis can be only performed by histopathologic studies [34]. In the case of the MRI T2 and PD WI low-intensity signal observed in transitional zone between left ventral and dorsal rat prostate lobes (Figure 8 and Figure 9) in histologic study was described as extensive necrosis, acute acini inflammation, and chronic stromal inflammation with fibrosis. In MRI T2 WI study (Figure 8) the loss of signal homogeneity of seminal vesicles was also characterized by zones with a low-intensity signal, in histologic study was diagnosed the development of a papillary adenoma (Figure 11C). Ultrasonography is very helpful for monitoring rat prostate dimension and form, and possibly to detect large lesions, and it is the only one that may be performed in awake animals. CT does not allow soft tissues differentiation (normal and cancerous) and is rarely used for human prostate cancer diagnosis or monitoring. CT in rats only detected the enlargement of prostate and seminal vesicles in prostate cancer animals and even the contrast-enhancement did not show ability to become evident prostate zones with distinct vascularization. However, CT allows a good especial prostate zonal localization and is frequently used in humans combined with PET (PET/CT) [37,44,49]. PET highlights the metabolic, molecular and cellular activity of prostate tissue. In humans, when used in combination with an imaging modality with a good anatomic information, such as CT or MRI, it allows prostate cancer diagnosis, staging, or detection of metastasis development [34]. However, the ideal radiotracer does not exist, and further research is required to find the optimal clinically helpful PET tracer in human prostate cancer [39]. The PET/CT studies in rat prostate cancer are scarce, and some works showed promising results in evaluation of prostate cancer therapies and hypoxia zones [44].

## 7. Conclusion

This work provides a description of the macroscopic and microscopic anatomy of the rat prostate, and its imaging by ultrasonography, CT and MRI. Both ultrasonography and MRI allow for a comprehensive and detailed study of the rat prostate and seminal vesicles, and are recommended for the study of prostate anatomy and for the monitoring of prostate diseases such as cancer. MRI show high sensitivity for detecting and delimiting abnormal prostatic tissue. The CT showed limitations in differentiation of internal prostate zones, eventually with different abnormal tissue even in contrast-enhancement studies.

## Figures and Tables

**Figure 1 diagnostics-09-00068-f001:**
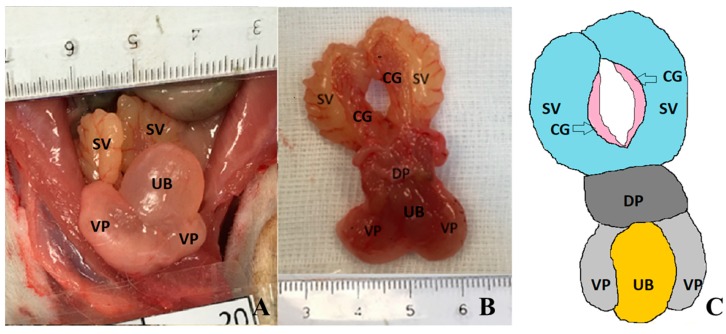
Macroscopic appearance of rat prostate and other surrounding anatomical structures at 61 weeks of age. (**A**) In situ photography of an animal from control group, ventral view; (**B**) Photography of prostate from an induced animal (dorsal view). Seminal vesicles and coagulative glands extended caudally. (**C**) Line diagram of prostate, urinary bladder and closed sex glands. Coagulating glands (CG), dorsal (DP) and ventral prostate lobes (VP), seminal vesicles (SV) and urinary bladder (UB).

**Figure 2 diagnostics-09-00068-f002:**
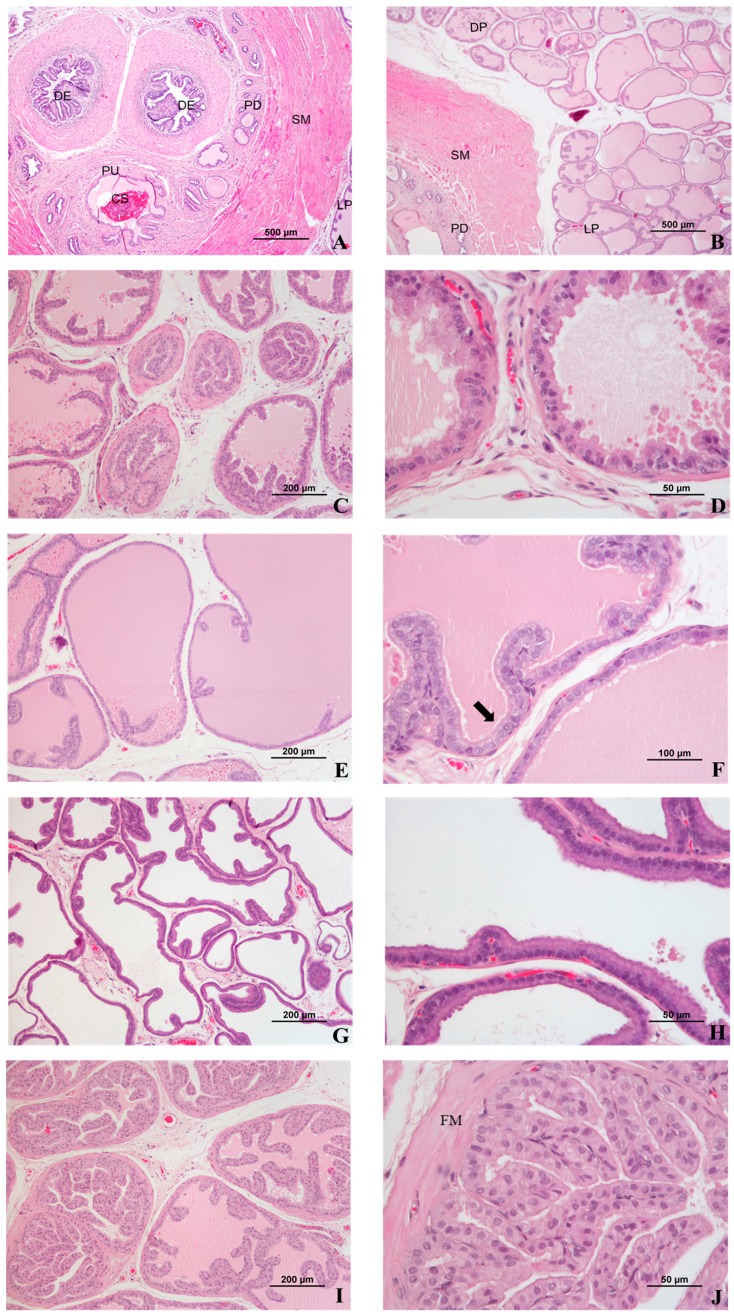
Microscopic features of normal rat prostate, 61 weeks of age. (**A**) Midtransverse section of the prostate showing prostatic urethra (PU) containing coagulated secretion (CS) (also denominated seminal plug or copulatory plug), and showing prostate ducts (PD) and ductus ejaculatorii (DE) enclosed by smooth muscle (SM) and surrounded by lateral prostate (LP). (**B**) Tissues surrounding PU with neighboring dorsal (DP) and lateral (LP) prostate lobes. (**C**) Dorsal prostate: small acini surrounded by a fibromuscular layer. (**D**) Dorsal prostate: cuboid to columnar cells presenting cytoplasmic blebs. (**E**) Lateral prostate showing moderate epithelial folding. (**F**) Lateral prostate acini lined by cuboid cell with supranuclear clear areas and brush border (arrow). (**G**) Ventral prostate showing acini with scarce infolding lined by basophilic cells. (**H**) Ventral prostate lined by columnar cells with basal nuclei and supranuclear cytoplasmic clear areas. (**I**) Coagulating gland showing tightly packed acini with intense infolding. (**J**) Coagulating gland presenting acini with high number of papillary structures, surrounded by a prominent fibromuscular layer (FM). (**A–J**) H&E staining.

**Figure 3 diagnostics-09-00068-f003:**
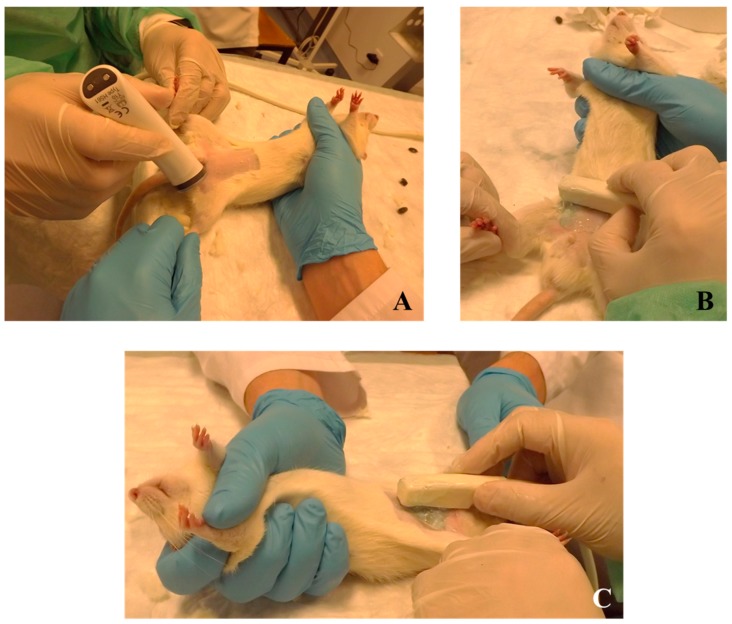
Preparation of a rat for a ultrasonographic exam. The animals should be restrained by a researcher and placed in supine position. The skin of the caudal aspect of the abdomen and the inguinal region should be shaved (**A**). Transverse (**B**) and longitudinal (**C**) scans of the prostate may be performed using B mode.

**Figure 4 diagnostics-09-00068-f004:**
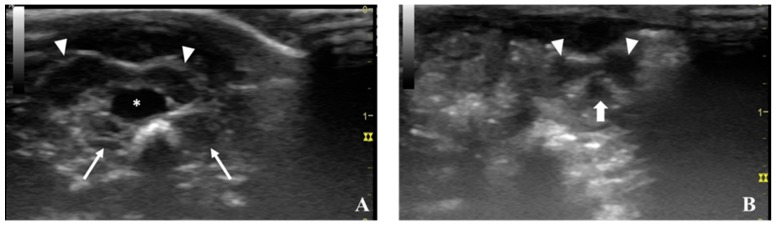
Transverse ultrasonographic scan of the prostate from control animals. The scan was performed in the caudal aspect of the abdomen (**A**) and in the inguinal region (**B**). In **A** can be observed the urinary bladder (asterisk) surrounded by the ventral lobes of the prostate (arrowheads) and the seminal vesicles (arrows), and in **B**, the neck of the urinary bladder surrounded by the ventral prostate lobes (arrowheads) and the dorsal prostate lobe (bold arrow). The 1 represents the depth (1 cm). The yellow II represents the focus of the ultrasound beam.

**Figure 5 diagnostics-09-00068-f005:**
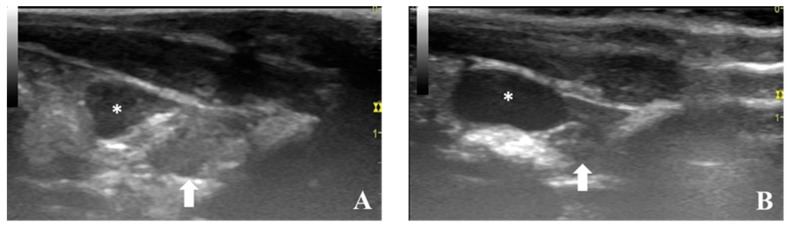
Longitudinal ultrasonographic scan of prostate from control animals (**A**,**B**). The dorsal prostate lobe (arrow) is observed dorsally to the neck of the urinary bladder (asterisk).

**Figure 6 diagnostics-09-00068-f006:**
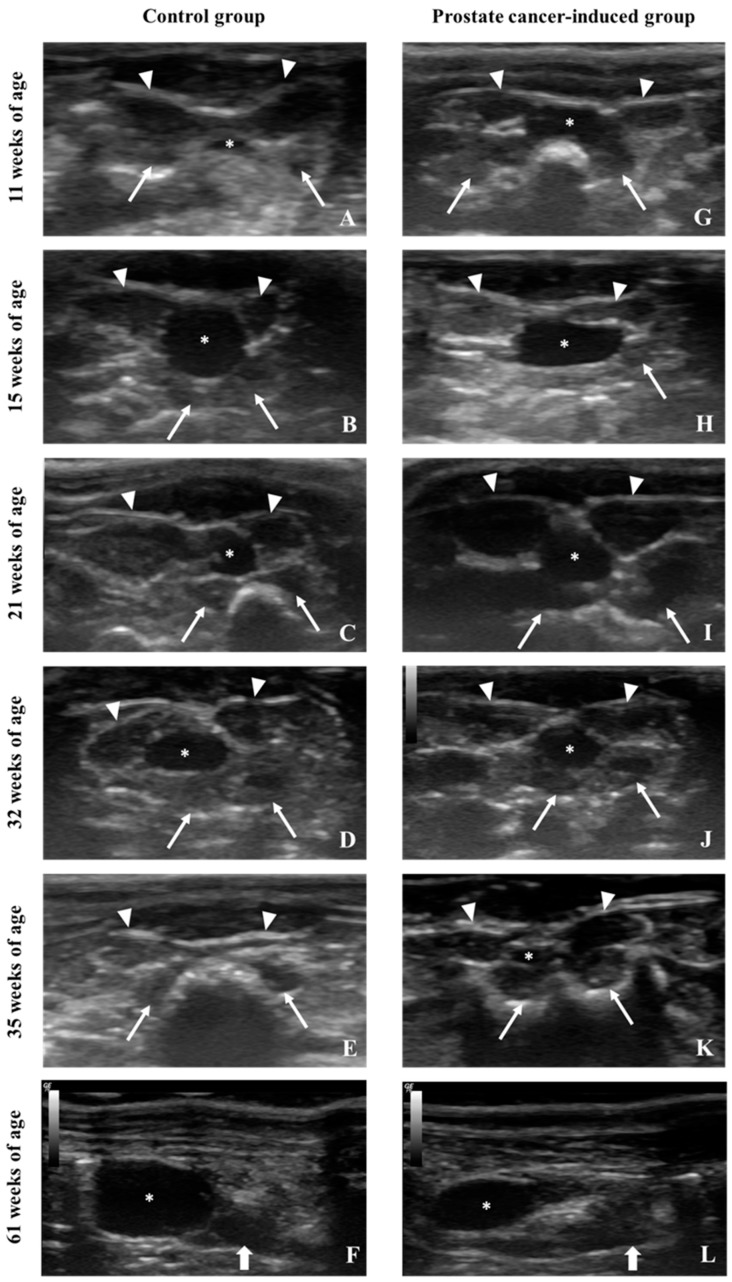
Prostates of both control (**A**–**F**) and prostate cancer-induced (**G**–**L**) animals were monitored by ultrasonography through the experimental protocol at the same time points. Transverse scans were performed at different abdomen levels one week before flutamide administration (**G**), three weeks after flutamide administration (**H**), six weeks after MNU administration (**I**), 17 weeks after MNU administration (**J**) and 20 weeks after MNU administration (**K**). Longitudinal scans were performed 46 weeks after MNU administration (**L**). Changes in prostate and seminal vesicles dimension are evident in images, as well as some alterations of internal echogenicity of these structures in cancer-induced animals. The urinary bladder (asterisk) is surrounded by the ventral prostate lobes (arrowheads), dorsal prostate (bold arrow) and seminal vesicles (arrows).

**Figure 7 diagnostics-09-00068-f007:**
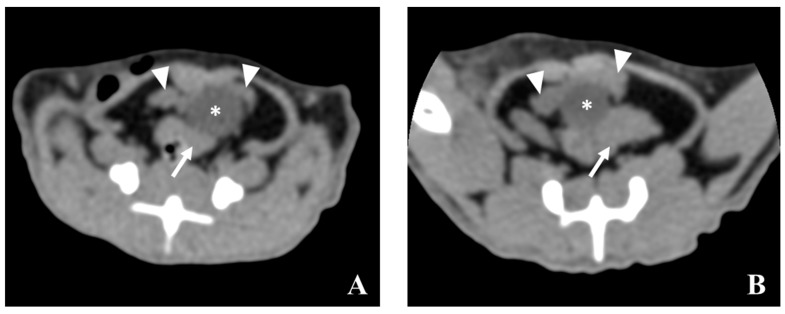
CT soft tissue window (WW 400, WL 40) transverse plane of prostate from control (**A**) and from cancer-induced (**B**) animals at 60 weeks of age without contrast. An enlargement of prostate and seminal vesicles from the cancer-induced animal is evident in the image (**B**). The urinary bladder (asterisk) is surrounded by the ventral prostate (arrowheads) and the seminal vesicles (arrow).

**Figure 8 diagnostics-09-00068-f008:**
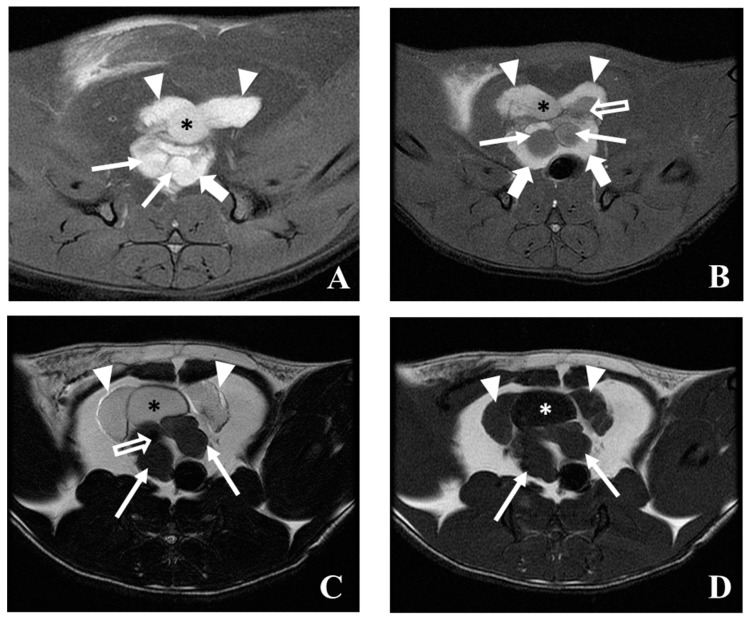
Transverse MRI scan at different levels of rat prostate at 58 weeks of age: (**A**) PD weighted image of a normal animal; (**B**) PD weighted image of a cancer-induced animal (similar level of A); (**C**) T2 weighted image of a cancer-induced animal; (**D**) T1 weighted image of a cancer-induced animal (similar level as (**C**)). The urinary bladder (asterisk) is surrounded by the ventral prostate (arrowheads), dorsal prostate (bold arrows) or seminal vesicles (arrows). A prostate and seminal vesicles enlargement is evident in cancer-induced animal, as well some changes with low-intensity signal of some zones of these structures (open arrows). In images (**A**) and (**B**), it is possible to observe the transition between the seminal vesicles and the dorsal prostate, standing out the loss of signal intensity of seminal vesicle in prostate cancer-induced animal.

**Figure 9 diagnostics-09-00068-f009:**
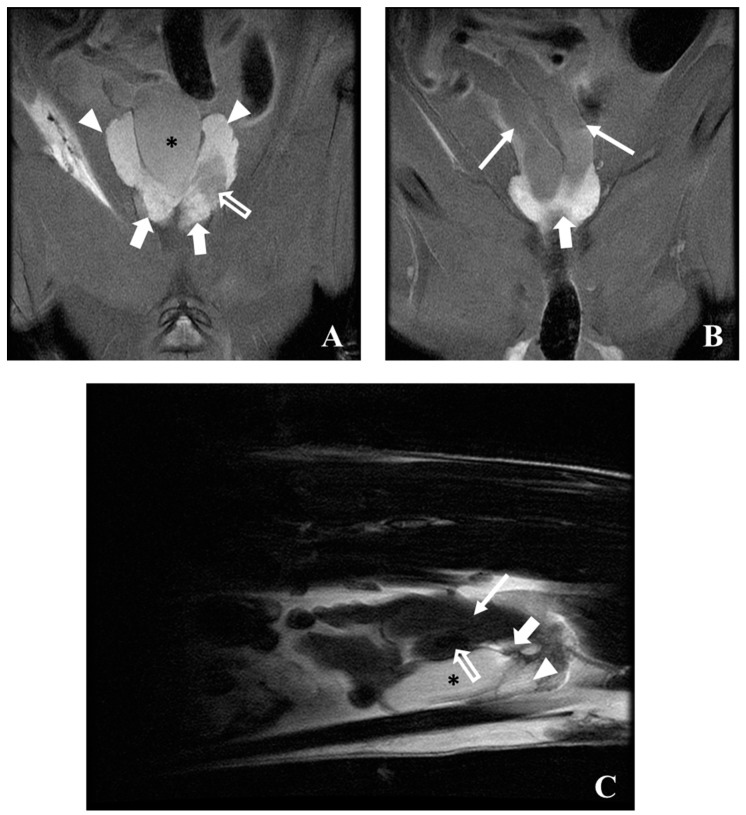
MRI PD weighted images of a cancer-induced animal, 58 weeks of age in dorsal (**A**,**B**) and sagittal (**C**) plane. The urinary bladder (asterisk) is well visible, surrounded by the dorsal (bold arrow) and ventral prostate (arrowhead), and the seminal vesicles (arrow). Low-intensity MRI signals were observed (open arrows) in transition dorsal/ventral prostate (**B**) and in seminal vesicles (**C**).

**Figure 10 diagnostics-09-00068-f010:**
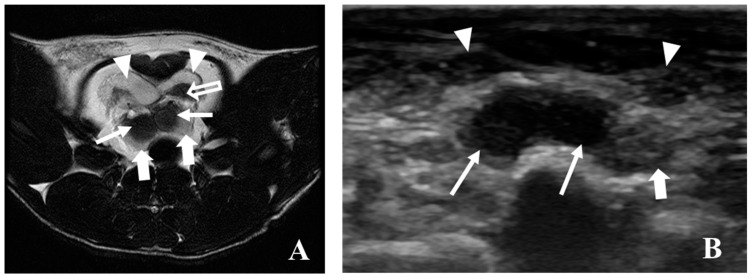
Comparison of an MRI T2 WI (**A**) and ultrasonographic (**B**) transverse scan of the rat prostate. In both images, it is possible to observe the ventral prostate (arrowheads) and the transition between the seminal vesicles (arrows) and the dorsal prostate (bold arrows). A low-intensity signal of prostate is observed in (**A**) (open arrow), no abnormal tissue echogenicity was evident in (**B**).

**Figure 11 diagnostics-09-00068-f011:**
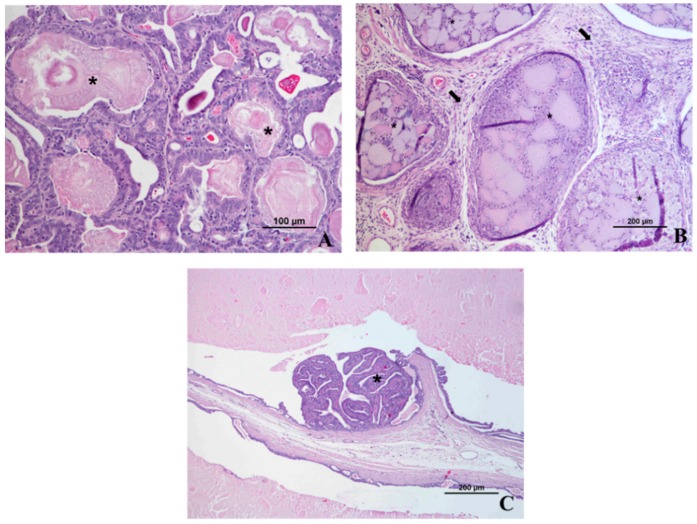
Microscopic appearance of abnormal rat seminal vesicles and prostate at 61 weeks of age in prostate cancer induced group. (**A**) Enlarged seminal vesicle showing corpora amylacea (*); (**B**) Dorsolateral left prostate lobe showing extensive necrosis, acute acini inflammation (*) and chronic stromal inflammation with fibrosis (arrow) (see normal dorsal and lateral prostate lobes in Figure 2B,F). (**C**) Seminal vesicle mural atrophy and presence of a papillary adenoma (*).

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
