# Peer review of "Anatomy and Imaging of Rat Prostate: Practical Monitoring in Experimental Cancer-Induced Protocols"

_diagnostics, 2019, doi:10.3390/diagnostics9030068_

Reviewer 1 Report

Dear editor,

In this manuscript the authors describe the anatomy of prostate of the rat, and also imaging modalities to visualize abnormalities of the prostate are described.

However, some  items are unclear for me in this manuscript and should be clarified:

- Many publications are available describing the anatomy of the rat. This manuscript does not really fill a gap.

- In the title imaging of the rat prostate is mantioned. However, a very important imaging technique, namely positon emission tomograph (PET) is completely missing in this manuscript. A lot of PETstudies are performed in (human) prostate at this moment. So at least this technique and results from prostate imaging PET studies should be discussed in this manuscript.

- A discussion about the value of different imaging techniques according to investigation of prostate diseases is missing. This discussion would be helpful to bring the manuscript in a broader perspective.

Author Response

Revision Note

First of all, thank you for the excellent review comments, they were very important for the improvement of the manuscript and the new manuscript version is undoubtably better. All the reviewers’ comments were analyzed, changes introduced in the text (highlighted in red color), and a detailed response is provided in this Revision Note:

Editor Comment

Your comment (YC): I find it a valuable contribution to the field. With regard to the comments from reviewers, my comments are: - Include PET: agree - fully reviewed and not only mention

Our answer (OA): The point “5.4. Positron emission tomography/Computed tomography (PET/CT)” was added.

YC: Confusion review/original: I find it OK and do not agree that it
should be moved to original contribution (to little own data), but
normal and OK to include own examples in a review.

OA: It was resubmitted as a review, the point 2. “Prostate study” was reformulated to avoid some confusion in this aspect.

YC: Besides, please confirm that whether all figures were drawn by yourself. If not, please confirm if figure permissions should be provided. If needed, please acquire from copyright holder and send scanned written or digital copy of permission to us via email.

OA: All the images are own.

Reviewer 1 

YC:  Many publications are available describing the anatomy of the rat. This manuscript does not really fill a gap. OA: The gap that we refer is not the normal prostate rat anatomy, but the imaging rat prostate cancer monitoring - the sentence was rewritten, and we think that now is better.

YC: In the title imaging of the rat prostate is mentioned. However, a very important imaging technique, namely positon emission tomograph (PET) is completely missing in this manuscript. A lot of PETstudies are performed in (human) prostate at this moment. So at least this technique and results from prostate imaging PET studies should be discussed in this manuscript. OA: We do not have PET/CT equipment so we didn`t have the possibility to perform this study in rats, but we added a point (5.4) to refer the use and importance of PET/CT in diagnosis of human prostate cancer.

YC:  A discussion about the value of different imaging techniques according to investigation of prostate diseases is missing. This discussion would be helpful to bring the manuscript in a broader perspective. OA: We added a discussion about imaging monitoring o rat prostate cancer

Reviewer 2

YC: This study was submitted as a review article. Since the authors present and mainly rely on own experimental work, it would be more appropriate to categorize the manuscript as original study. To do this, the authors are requested to restructure the manuscript following the author instructions for original research articles using the diagnostics template provided. OA: This manuscript was resubmitted as a review. However, to avoid eventual confusion review/original the point 2. “Prostate study” and other work aspects were reformulated.

YC: The aim of the review was “to provide a complete and systematic overview of rat prostate anatomy and imaging”. Unfortunately, the anatomy of the rat prostate was not fully illustrated, requiring the consultation of references, which are mostly not readily available to the reader. The review could be significantly improved by illustrating the major aspects of rat prostate anatomy either as sketches or even better in combination with micrographs. OA: Figure 1 was reformulated, and additional annotations were made in the images.

YC: I miss the “practical guidelines” announced in the title of the manuscript. The authors should add work flows for demonstration. OA: The title was changed, and the word guidelines was deleted.

YC: Finally, the authors recommend US, CT and MRI imaging techniques for monitoring prostate cancer in the rat. However, they did not show a positive or suspect area in either of their images. OA: Additional annotations were placed in images for this purpose.

Detailed comments:

YC: Line 53: Is it correct that the authors investigated only “three” prostates? This statement seems to be in conflict with the description below (line 64). Please report the exact group sizes of the prostate cancer and the control group. OA: The text was rewritten, and the number of used animals mentioned (2 normal and 3 with prostate cancer).

YC: The authors should closely refer to figure 2 when describing the microscopic anatomy of the prostate. They should also declare if the micrographs were taken from an animal from the prostate cancer or from the control group. Always report the age of the animal in the legend. OA: This information was added.

YC: Fig. 1: I recommend including a macrograph or sketch showing the situation in situ including the urinary bladder and the urethra. The coagulation gland should also be depicted. OA: Fig. 1 was changed, and your suggestions added.

YC: Line 99: The authors should specify the necessary sectioning in a sketch and by histological micrographs. Otherwise, their statement remains theoretical and incomprehensible. OA: Micrographs legend and text were changed, we think that is better now.

YC: Fig. 2: The position of the colliculus seminalis remains unclear. It should be demonstrated in a transversal section of the urethra. OA: Some images of Figure 2 and the legends were changed, we think that is better now.

YC: Line 155: the paragraph “5. Prostate imaging” is rudimentary and does not reveal much information. It should be integrated as introduction to the following explanation of the different imaging modalities. OA: We agree, the structure of this work section changed.

YC: Duplicate numbering: line 161 “6. Ultrasonography” and line 242 “6. Computed tomography (CT)” OA: It was changed.

YC: Fig. 6: the respective time points of US should be included as labels in the images for better readability. OA: The information was added.

YC: Fig. 6: I cannot comprehend the differences in size. Please indicate by labelling the maximum diameters. OA: We do not introduce this change, we try but figures became very overloaded with information and the general aspect was lost a little.

YC: The legend of Fig. 6 is confusing. E.g. “…, three weeks after flutamide administration (15 weeks of age) (B and H)”. Is this correct? (B) is the control and should not have received flutamide. OA: Legend was reformulated.

YC: Line 257: The author state that contrast-enhancement showed no improvement, but they do not present any evidence. Are the CT images shown in Fig. 7 done with or without Ultravist 300? Labelling of the tumor is missing. OA: Legend was reformulated. Distinct opacity was not visible in any prostate zone.

YC: The authors should finally judge the use of the CT for the detection of prostate cancer in the rat. OA: CT limitations and its use in humans was added to the text.

YC: Fig. 8: Seminal vesicles are not visible in (A). Use scan equivalent to (B). OA: We replace the Fig. A by other more caudal, the MRI T2 and DP signal of SV changed in control and prostate cancer animals, this information was added.

YC: Final evaluation of the different imaging modalities to detect and measure the size of induced prostate cancer in the rat is missing. OA: This information was added to the discussion.

Reviewer 2 Report

This study was submitted as a review article. Since the authors present and mainly rely on own experimental work, it would be more appropriate to categorize the manuscript as original study. To do this, the authors are requested to restructure the manuscript following the author instructions for original research articles using the diagnostics template provided.

The aim of the review was “to provide a complete and systematic overview of rat prostate anatomy and imaging”. Unfortunately, the anatomy of the rat prostate was not fully illustrated, requiring the consultation of references, which are mostly not readily available to the reader. The review could be significantly improved by illustrating the major aspects of rat prostate anatomy either as sketches or even better in combination with micrographs.

I miss the “practical guidelines” announced in the title of the manuscript. The authors should add work flows for demonstration.

Finally, the authors recommend US, CT and MRI imaging techniques for monitoring prostate cancer in the rat. However, they did not show a positive or suspect area in either of their images.  

Detailed comments:

Line 53: Is it correct that the authors investigated only “three” prostates? This statement seems to be in conflict with the description below (line 64). Please report the exact group sizes of the prostate cancer and the control group.

The authors should closely refer to figure 2 when describing the microscopic anatomy of the prostate. They should also declare if the micrographs were taken from an animal from the prostate cancer or from the control group. Always report the age of the animal in the legend.

Fig. 1: I recommend including a macrograph or sketch showing the situation in situ including the urinary bladder and the urethra. The coagulation gland should also be depicted.

Line 99: The authors should specify the necessary sectioning in a sketch and by histological micrographs. Otherwise, their statement remains theoretical and incomprehensible.

Fig. 2: The position of the colliculus seminalis remains unclear. It should be demonstrated in a transversal section of the urethra.

Line 155: the paragraph “5. Prostate imaging” is rudimentary and does not reveal much information. It should be integrated as introduction to the following explanation of the different imaging modalities.

Duplicate numbering: line 161 “6. Ultrasonography” and line 242 “6. Computed tomography (CT)”

Fig. 6: the respective time points of US should be included as labels in the images for better readability.

Fig. 6: I cannot comprehend the differences in size. Please indicate by labelling the maximum diameters.

The legend of Fig. 6 is confusing. E.g. “…, three weeks after flutamide administration (15 weeks of age) (B and H)”. Is this correct? (B) is the control and should not have received flutamide.

Line 257: The author state that contrast-enhancement showed no improvement, but they do not present any evidence. Are the CT images shown in Fig. 7 done with or without Ultravist 300? Labelling of the tumor is missing.

The authors should finally judge the use of the CT for the detection of prostate cancer in the rat.

Fig. 8: Seminal vesicles are not visible in (A). Use scan equivalent to (B).

Final evaluation of the different imaging modalities to detect and measure the size of induced prostate cancer in the rat is missing.

Author Response

Revision Note

First of all, thank you for the excellent review comments, they were very important for the improvement of the manuscript and the new manuscript version is undoubtably better. All the reviewers’ comments were analyzed, changes introduced in the text (highlighted in red color), and a detailed response is provided in this Revision Note:

Editor Comment

Your comment (YC): I find it a valuable contribution to the field. With regard to the comments from reviewers, my comments are: - Include PET: agree - fully reviewed and not only mention

Our answer (OA): The point “5.4. Positron emission tomography/Computed tomography (PET/CT)” was added.

YC: Confusion review/original: I find it OK and do not agree that it
should be moved to original contribution (to little own data), but
normal and OK to include own examples in a review.

OA: It was resubmitted as a review, the point 2. “Prostate study” was reformulated to avoid some confusion in this aspect.

YC: Besides, please confirm that whether all figures were drawn by yourself. If not, please confirm if figure permissions should be provided. If needed, please acquire from copyright holder and send scanned written or digital copy of permission to us via email.

OA: All the images are own.

Reviewer 1 

YC:  Many publications are available describing the anatomy of the rat. This manuscript does not really fill a gap. OA: The gap that we refer is not the normal prostate rat anatomy, but the imaging rat prostate cancer monitoring - the sentence was rewritten, and we think that now is better.

YC: In the title imaging of the rat prostate is mentioned. However, a very important imaging technique, namely positon emission tomograph (PET) is completely missing in this manuscript. A lot of PETstudies are performed in (human) prostate at this moment. So at least this technique and results from prostate imaging PET studies should be discussed in this manuscript. OA: We do not have PET/CT equipment so we didn`t have the possibility to perform this study in rats, but we added a point (5.4) to refer the use and importance of PET/CT in diagnosis of human prostate cancer.

YC:  A discussion about the value of different imaging techniques according to investigation of prostate diseases is missing. This discussion would be helpful to bring the manuscript in a broader perspective. OA: We added a discussion about imaging monitoring o rat prostate cancer

Reviewer 2

YC: This study was submitted as a review article. Since the authors present and mainly rely on own experimental work, it would be more appropriate to categorize the manuscript as original study. To do this, the authors are requested to restructure the manuscript following the author instructions for original research articles using the diagnostics template provided. OA: This manuscript was resubmitted as a review. However, to avoid eventual confusion review/original the point 2. “Prostate study” and other work aspects were reformulated.

YC: The aim of the review was “to provide a complete and systematic overview of rat prostate anatomy and imaging”. Unfortunately, the anatomy of the rat prostate was not fully illustrated, requiring the consultation of references, which are mostly not readily available to the reader. The review could be significantly improved by illustrating the major aspects of rat prostate anatomy either as sketches or even better in combination with micrographs. OA: Figure 1 was reformulated, and additional annotations were made in the images.

YC: I miss the “practical guidelines” announced in the title of the manuscript. The authors should add work flows for demonstration. OA: The title was changed, and the word guidelines was deleted.

YC: Finally, the authors recommend US, CT and MRI imaging techniques for monitoring prostate cancer in the rat. However, they did not show a positive or suspect area in either of their images. OA: Additional annotations were placed in images for this purpose.

Detailed comments:

YC: Line 53: Is it correct that the authors investigated only “three” prostates? This statement seems to be in conflict with the description below (line 64). Please report the exact group sizes of the prostate cancer and the control group. OA: The text was rewritten, and the number of used animals mentioned (2 normal and 3 with prostate cancer).

YC: The authors should closely refer to figure 2 when describing the microscopic anatomy of the prostate. They should also declare if the micrographs were taken from an animal from the prostate cancer or from the control group. Always report the age of the animal in the legend. OA: This information was added.

YC: Fig. 1: I recommend including a macrograph or sketch showing the situation in situ including the urinary bladder and the urethra. The coagulation gland should also be depicted. OA: Fig. 1 was changed, and your suggestions added.

YC: Line 99: The authors should specify the necessary sectioning in a sketch and by histological micrographs. Otherwise, their statement remains theoretical and incomprehensible. OA: Micrographs legend and text were changed, we think that is better now.

YC: Fig. 2: The position of the colliculus seminalis remains unclear. It should be demonstrated in a transversal section of the urethra. OA: Some images of Figure 2 and the legends were changed, we think that is better now.

YC: Line 155: the paragraph “5. Prostate imaging” is rudimentary and does not reveal much information. It should be integrated as introduction to the following explanation of the different imaging modalities. OA: We agree, the structure of this work section changed.

YC: Duplicate numbering: line 161 “6. Ultrasonography” and line 242 “6. Computed tomography (CT)” OA: It was changed.

YC: Fig. 6: the respective time points of US should be included as labels in the images for better readability. OA: The information was added.

YC: Fig. 6: I cannot comprehend the differences in size. Please indicate by labelling the maximum diameters. OA: We do not introduce this change, we try but figures became very overloaded with information and the general aspect was lost a little.

YC: The legend of Fig. 6 is confusing. E.g. “…, three weeks after flutamide administration (15 weeks of age) (B and H)”. Is this correct? (B) is the control and should not have received flutamide. OA: Legend was reformulated.

YC: Line 257: The author state that contrast-enhancement showed no improvement, but they do not present any evidence. Are the CT images shown in Fig. 7 done with or without Ultravist 300? Labelling of the tumor is missing. OA: Legend was reformulated. Distinct opacity was not visible in any prostate zone.

YC: The authors should finally judge the use of the CT for the detection of prostate cancer in the rat. OA: CT limitations and its use in humans was added to the text.

YC: Fig. 8: Seminal vesicles are not visible in (A). Use scan equivalent to (B). OA: We replace the Fig. A by other more caudal, the MRI T2 and DP signal of SV changed in control and prostate cancer animals, this information was added.

YC: Final evaluation of the different imaging modalities to detect and measure the size of induced prostate cancer in the rat is missing. OA: This information was added to the discussion.

Round  2

Reviewer 1 Report

I mentioned to review PET/CT imaging as a very important technique for prostate imaging. However, this discussion is very limited. Only FDG and choline are shorly mentioned. Several other important radiotracters like PSMA tracers are not mentioned or discussed at all. As a result, the PET/CT imaging is not brought into a broader perspective sufficiently by comparing PET/CT to other techniques for prostate imaging, as I asked for.

In conclusion, to my opinion the authors did not fully described my comments.

Author Response

Revision Note

First of all, thank you for the excellent review comments. All the reviewers’ comments were analyzed, changes introduced in the text, and a detailed response is in this revision Note:

Reviewer 1

Your comment (YC): I mentioned to review PET/CT imaging as a very important technique for prostate imaging. However, this discussion is very limited. Only FDG and choline are shortly mentioned. Several other important radiotracters like PSMA tracers are not mentioned or discussed at all. As a result, the PET/CT imaging is not brought into a broader perspective sufficiently by comparing PET/CT to other techniques for prostate imaging, as I asked for.
In conclusion, to my opinion the authors did not fully described my comments.  

Our answer (OA): PET/CT imaging was enlarged and PSMA included.

Reviewer 2 Report

The authors rewrote significant parts of the manuscript, and also added an extensive Discussion. They also added Fig. 11 showing the histological alterations in the cancer induced group. I recommend they add within the legend a cross reference to the corresponding normal prostate histology in Fig. 2.  

In total the manuscript was greatly improved; it is well written, and does add to literature.

I have a few further comments:

YC: Fig. 2: The position of the colliculus seminalis remains unclear. It should be demonstrated in a transversal section of the urethra. OA: Some images of Figure 2 and the legends were changed, we think that is better now.

RE Fig. 2A: the authors labeled the ducts as VD = Vas deferens. However, the ducts within the prostate which empty into the urethra via the colliculus seminals should be termed ductus ejaculatorii.

Line 319: "Unlike CT, MRI does not use ionizing radiation and works by changing the position of atoms..."
This sentence is misleading. MRI does not change the position of atoms but measures the reversal of the atomic spin. Please rephrase.

Fig. 8: C (T2) and D (T1) seem to be the same image differently weighted. The authors should indicate in the legend.

Author Response

Revision Note

First of all, thank you for the excellent review comments. All the reviewers’ comments were analyzed, changes introduced in the text, and a detailed response is in this revision Note:

Reviewer 2

Your Comments(YO): The authors rewrote significant parts of the manuscript, and also added an extensive Discussion. They also added Fig. 11 showing the histological alterations in the cancer induced group. I recommend they add within the legend a cross reference to the corresponding normal prostate histology in Fig. 2.  

Our answer (OA): Added.

YC: In total the manuscript was greatly improved; it is well written, and does add to literature. OA: Thank you for your positive feedback.

YC: Fig. 2: The position of the colliculus seminalis remains unclear. It should be demonstrated in a transversal section of the urethra. 

OA: In the first version of the manuscript colliculus seminalis was poorly identified, but in the present version the figure is not adequate for this purpose and the authors don´t intend to show it, so it is not mentioned in figure 2.

YC: OA: Some images of Figure 2 and the legends were changed, we think that is better now. RE Fig. 2A: the authors labeled the ducts as VD = Vas deferens. However, the ducts within the prostate which empty into the urethra via the colliculus seminals should be termed ductus ejaculatorii. OA: Changed.

YC: Line 319: "Unlike CT, MRI does not use ionizing radiation and works by changing the position of atoms..." This sentence is misleading. MRI does not change the position of atoms but measures the reversal of the atomic spin. Please rephrase. 

OA: Rephrased.

YC: Fig. 8: C (T2) and D (T1) seem to be the same image differently weighted. The authors should indicate in the legend. 

OA: Indicated.

Round  3

Reviewer 1 Report

No comments.

Author Response

Thank you for your comments. We performed the requested corrections.